# A Systematic Review and Meta-Analysis of Supramarginal Resection versus Gross Total Resection in Glioblastoma: Can We Enhance Progression-Free Survival Time and Preserve Postoperative Safety?

**DOI:** 10.3390/cancers15061772

**Published:** 2023-03-15

**Authors:** Johannes Wach, Martin Vychopen, Andreas Kühnapfel, Clemens Seidel, Erdem Güresir

**Affiliations:** 1Department of Neurosurgery, University Hospital Leipzig, 04103 Leipzig, Germany; 2Institute for Medical Informatics, Statistics and Epidemiology, Leipzig University, 04107 Leipzig, Germany; 3Department of Radiation Oncology, University Hospital Leipzig, 04103 Leipzig, Germany

**Keywords:** extent of resection, glioblastoma, meta-analysis, progression-free survival, supramarginal resection

## Abstract

**Simple Summary:**

Glioblastoma tumor cells are frequently found in areas distant from conventional imaging abnormalities. These cells are thought to play a role in tumor progression after surgery of the initially defined tumor region on MRI. Supramarginal resection (SMR) is an emerging technique in neuro-oncological surgery that may improve tumor control. However, the impact of SMR on PFS and postoperative complications has not been investigated so far. This study performed the first systematic review and meta-analysis of the literature on SMR and further investigated the impact of SMR on PFS and postoperative complications. The results of this study suggest that while the current evidence is of low quality, SMR may improve PFS without affecting postoperative surgical complications. However, prospective research with larger patient cohorts and clearly defined SMR techniques is needed to confirm these findings.

**Abstract:**

To date, gross total resection (GTR) of the contrast-enhancing area of glioblastoma (GB) is the benchmark treatment regarding surgical therapy. However, GB infiltrates beyond those margins, and most tumors recur in close proximity to the initial resection margin. It is unclear whether a supramarginal resection (SMR) enhances progression-free survival (PFS) time without increasing the incidence of postoperative surgical complications. The aim of the present meta-analysis was to investigate SMR with regard to PFS and postoperative surgical complications. We searched for eligible studies comparing SMR techniques with conventional GTR in PubMed, Cochrane Library, Web of Science, and Medline databases. From 3158 initially identified records, 11 articles met the criteria and were included in our meta-analysis. Our results illustrate significantly prolonged PFS time in SMR compared with GTR (HR: 11.16; 95% CI: 3.07–40.52, *p* = 0.0002). The median PFS of the SMR arm was 8.44 months (95% CI: 5.18–11.70, *p* < 0.00001) longer than the GTR arm. The rate of postoperative surgical complications (meningitis, intracranial hemorrhage, and CSF leaks) did not differ between the SMR group and the GTR group. SMR resulted in longer median progression-free survival without a negative postoperative surgical risk profile. Multicentric prospective randomized trials with a standardized definition of SMR and analysis of neurologic functioning and health-related quality of life are justified and needed to improve the level of evidence.

## 1. Introduction

Glioblastoma (GB) is the most common and aggressive primary malignant central nervous system tumor [1]. Despite maximum cytoreductive resection and adjuvant radiochemotherapy, survival time often remains poor [2]. Even in recent randomized phase 3 trials involving innovative strategies such as immune-checkpoint inhibition or antibody–drug conjugates, the median progression-free survival remains at 6 months in the largest subgroup of GB with unmethylated MGMT promotors [3,4].

Although it is known that glioma cells can be found throughout the brain, up to 80% of patients develop initial tumor recurrence in close proximity to the resection site (https://doi.org/10.1016/0360-3016(89)90941-3 (accessed on 1 February 2023)) [5,6,7]. Reasons for this might be manifold, e.g., the growth-promoting influence of peri-tumoral microenvironments, local glioma regrowth with help of functional tumor cell networks, or simply higher tumor cell density around the resection cavity [8,9,10].

Despite the high local relapse rate, maximum cytoreductive surgery has been shown to prolong survival [11,12]. There is currently an emerging debate regarding the benefit of resection beyond the classical resection margin (“supramarginal or supra-total resection”) on overall survival (OS) and progression-free survival (PFS) time [13]. While aggressive surgical approaches are essential regarding tumor control, preserving neurological function and avoiding complications should remain the second priority of surgery [14]. A recent meta-analysis found a moderate OS benefit for patients who underwent a supramarginal resection (SMR) [15]. Tumor progression is a hallmark in GB because recurrent GBs are driven by intrinsic (e.g., MGMT upregulation and increased tumor mutation burden) and extrinsic (e.g., hypoxia and immuno-suppressive tumor microenvironments) mechanisms developing resistance to therapies [16]. Furthermore, GB progression and postoperative complications are also significantly associated with negative changes in patient-reported health-related quality of life [17,18]. To date, the impact of supramarginal resection (SMR) compared to gross total resection (GTR) on progression-free survival and postoperative complications (e.g., mortality, meningitis, intracranial hemorrhage, and cerebrospinal fluid (CSF) leaks) has not yet been investigated in a meta-analysis.

The present systematic review and meta-analysis aim to investigate supramarginal resection compared with gross total resection regarding the probability of progression-free survival and perioperative surgical complications.

## 2. Materials and Methods

In this meta-analysis, the authors strictly followed the PRISMA checklist (see Appendix A) [19] and the *Cochrane Handbook for Systematic Reviews of Interventions* version 6.3 [20]. The study was registered in the “*International Prospective Register of Systematic Reviews*” (PROSPERO) in 2023 (CRD42023395933), and the detailed prespecified protocol is available upon request.

### 2.1. Inclusion and Exclusion Criteria

The authors performed a systematic search in November, 2022 of the PubMed, Cochrane Library, Web of Science, and Medline databases using the search terms “glioblastoma”, “supratotal resection”, “supramaximal resection”, “supracomplete resection”, “FLAIR resection”, “lobectomy”, and “supramarginal resection”. The search was limited to “human studies”, “clinical trials”, and “English” language publications, with a literature search that included all results up to 31 October 2022. The inclusion criteria were formulated using the PICOS (population, intervention, comparator, outcomes, and study design) framework [21], with the following criteria: patients had undergone treatment for GB; relevant surgical resections were performed; SMR results were compared to conventional GTR regarding PFS or perioperative complications; all prespecified endpoints were reported; and the studies were comparative studies comparing different surgical resection techniques. Records such as reviews, study protocols, letters, conference abstracts, unpublished papers, animal experiments, and studies with insufficient data (e.g., no description of surgical resection technique with definition of extent of resection) were excluded. Previous meta-analyses and reviews were also searched for studies matching the inclusion and exclusion criteria.

The identified articles were further examined in a stepwise workflow that involved screening titles of the studies, abstracts, and full texts independently by two authors (M.V. and J.W.), with any disagreement settled by a third author (E.G.).

### 2.2. Data Extraction and Clinical Endpoints

Study names, first authors, year of publication, country, study design, level of evidence, number of centers (mono-, bi-, or multi-centric), and other relevant data were extracted as baseline data. Definitions of SMR, demographics (sex and age), and duration of PFS follow-up were extracted. Furthermore, statistical methods regarding the analysis of PFS were recorded. The following perioperative complications and outcomes were recorded from the identified studies: postoperative Karnofsky performance status (KPS), postoperative new neurological deficits, mortality, postoperative intracranial hemorrhage necessitating revision surgery, postoperative meningitis, and postoperative CSF leaks necessitating medical treatment.

### 2.3. Statistics

The meta-analyses were conducted using Review Manager Web (RevMan Web version 5.4.1 from The Cochrane Collaboration). The “Generic inverse variance” method was used for statistical analysis, whereby a pooled hazard ratio (HR) was determined from the natural logarithm (LN) of the individual HR (LN (HR)) and the corresponding 95% confidence interval (CI). For the hazard ratio (HR), the standard error (SE) for the LN (OR) was calculated from the 95% CI using the following formula: SE = (LN (upper CI limit) − LN (lower CI limit))/3.92 (according to the *Cochrane Handbook for Systematic Reviews of Interventions* version 6.3 [20]). Standard deviations were obtained from the 95% CI limits. Statistical heterogeneity and inconsistency were investigated using x^2^ and I^2^ statistics, respectively, where an I^2^ value of 50% or more indicated substantial heterogeneity [22]. The weight of the individual studies’ relative contribution, based on the sample size, was taken into consideration for estimating treatment effects. Random effect models were used to generate forest plots displaying the pooled estimates [22]. Publication bias was assessed using two methods: (1) visually examining funnel plots of included studies, and (2) performing Begg’s tests to evaluate the data’s asymmetry [23]. Begg’s tests were conducted using MedCalc (version 20.123 for Windows), with a *p*-value < 0.05 considered indicative of bias. Pooled OR and pooled HR estimates were used to express the effect sizes, and the following endpoints were investigated: progression-free survival, postoperative meningitis, postoperative intracranial hemorrhage, and postoperative CSF leaks.

## 3. Results

### 3.1. Literature Search

According to the defined search algorithm (see Section 2.1 and Section 2.2), a total of 3158 articles were initially identified (see Figure 1). After a review of the study titles, abstracts, and full texts, 3147 articles were excluded, leaving 11 articles eligible for the meta-analysis. These 11 articles involved a total of 1168 patients. A total of 8 of the 11 articles provided data regarding PFS, whereas 7 of the 11 articles provided data regarding perioperative complications.

### 3.2. Characteristics of Included Studies Regarding the Analysis of Progression-Free Survival

The included studies of the present systematic review regarding progression-free survival were published between 2013 and 2020. Seven records reported results from retrospective studies [24,25,26,27,28,29,30], whereas only one study provided data from a prospective study [31]. The summary of the major key characteristics of all included studies is provided in Table 1. For further information regarding the number of patients, level of evidence, demographics, and duration of follow-up in the included studies, see Table 1. After tumor resection, conventional radiochemotherapy with temozolomide was the standard treatment in all studies.

The details of SMR techniques, PFS times in supramarginally and gross totally resected GB patients, statistical methods, and duration of follow-up are summarized in Table 2. Generally, the following four SMR techniques were identified: resection with a margin of at least 1 cm surrounding the gadolinium-enhancing tumor in the normal white matter and overlying cortex [24,25], lobectomy [26,28,29], resection of the fluid-attenuated inversion recovery (FLAIR) signal alterations if deemed possible [27,30], and resection of the involved entire gyrus [31]. All studies, except for the study by Hamada et al. [31], performed log-rank tests comparing the probability of progression-free survival. All studies, except for the studies by Mampre et al. [30] and De Bonis et al. [25], found that the SMR techniques were superior to conventional GTR. However, only Glenn et al. [24] and Roh et al. [28] provide complete results from a multivariate Cox regression analysis, including hazard ratios, 95% confidence intervals, and p-values. Further studies reported only partial results of the Cox regression analysis [27,29], performed multivariate logistic regression analysis [26], or performed no multivariate analysis of PFS [25,31].

### 3.3. Impact of Supramarginal Resection on Progression-Free Survival in Glioblastoma

Two studies reported complete multivariate Cox regression data on PFS. Fifty-six patients were allocated into either the supramarginal resection arm or the gross total resection arm (27 vs. 29). Pooling of the results showed a significant association between supramarginal resection and progression-free survival (HR: 11.16; 95% CI: 3.07–40.52, *p* = 0.0002). Figure 2 shows a forest plot displaying the results of the analysis. No significant heterogeneity was present (I^2^ = 0%, *p* = 0.97).

Further analysis of the median PFS times (months) in patients (*n* = 332) who underwent SMR (*n* = 130) or GTR (*n* = 202) was performed. All studies, except for the studies by Hamada et al. [31] and Mampre et al. [30], were included in this analysis. The study by Hamada et al. [31] was excluded because PFS times were only reported stratified by the anatomical location (frontal, temporal, occipital, and parietal). Furthermore, Mampre et al. [30] only reported the overall median time to tumor progression. Compared with GTR, SMR of GB resulted in a longer PFS of 8.44 months (95% CI: 5.18–11.70, *p* < 0.00001). No significant heterogeneity was present (I^2^ = 34%, *p* = 0.18). Figure 3 summarizes those results.

### 3.4. Characteristics of Included Studies Regarding the Analysis of Perioperative Complications

The included studies of the present systematic review regarding perioperative complications were published between 2015 and 2020. The summary of major key characteristics of all included studies is provided in Table 3. No events of perioperative mortality within 30 days following surgery were reported in any trial. For further information regarding postoperative meningitis, postoperative intracranial hemorrhage, and postoperative CSF leaks in the included studies, see Table 3.

The studies were further reviewed regarding postoperative KPS and new neurological deficits. Appendix A summarizes the current data regarding postoperative KPS and new neurological deficits in studies comparing SMR with conventional GTR. Postoperative KPS values are provided in only six studies [25,26,28,29,33,34], and only four [26,28,29,33] of those studies further stratified the postoperative KPS values by EOR. The study by Schneider et al. [26] revealed that patients who underwent a temporal lobectomy had a significantly superior KPS at 12 months after surgery compared with patients who underwent a conventional GTR. The other three studies comparing postoperative KPS values stratified by EOR found no significant differences [28,29,33]. Statistical analysis of the postoperative KPS was not possible because different statistical values (e.g., mean or median values with ranges or interquartile ranges) at different time points are given. The rates of new postoperative neurological deficits are described in seven studies [24,27,30,31,32,33,34], and three of those studies describe the incidences of new postoperative deficits in patients who underwent either SMR or GTR [24,27,33]. No significant associations between SMR or GTR with the onset of new postoperative neurological deficits were found. Statistical analysis of new postoperative neurological deficits was not possible because only one event in the study by Glenn et al. [24] was observed, and the other studies comparing new neurological deficits stratified by SMR or GTR observed no events [27,33].

### 3.5. Impact of Supramarginal Resection on Postoperative Complications

#### 3.5.1. Postoperative Meningitis

Three studies reported data regarding the incidence of postoperative meningitis in patients who underwent either SMR or GTR. One-hundred-and-seventy-two patients were allocated into either the SMR arm or the GTR arm (62 vs. 110). Three patients (3/62; 4.8%) in the SMR arm suffered from postoperative meningitis, whereas one patient had postoperative meningitis in the GTR arm (1/110; 0.9%). Pooling of the results revealed no significant difference between SMR and GTR with regard to the incidence of postoperative meningitis (OR: 4.20; 95% CI: 0.70–25.18, *p* = 0.12). Figure 4 shows a forest plot displaying the results of the analysis. (No significant heterogeneity was present (I^2^ = 0%, *p* = 0.95)).

#### 3.5.2. Postoperative Intracranial Hemorrhage

Two studies reported data regarding the incidence of postoperative intracranial hemorrhage in patients who underwent either SMR or GTR. Eight-hundred-and-ninety-two patients were allocated into either the SMR arm or the GTR arm (650 vs. 242). Nine patients (9/650; 1.4%) in the SMR arm suffered from postoperative intracranial hemorrhage, and two patients had a postoperative intracranial hemorrhage in the GTR arm (2/242; 0.8%). Pooling of the results revealed no significant difference between SMR and GTR with regard to the incidence of postoperative intracranial hemorrhage (OR: 1.69; 95% CI: 0.24–12.05, *p* = 0.60). Figure 5 shows a forest plot displaying the results of the analysis. (No significant heterogeneity was present (I^2^ = 14%, *p* = 0.28)).

#### 3.5.3. Postoperative CSF Leaks

Two studies reported data regarding the incidence of postoperative intracranial hemorrhage in patients who underwent either SMR or GTR. One-hundred-and-thirty patients were allocated into either the SMR arm or the GTR arm (52 vs. 78). One patient (1/52; 1.9%) in the SMR arm suffered from a postoperative CSF leak, whereas one patient had a postoperative CSF leak in the GTR arm (1/78; 1.3%). Pooling of the results revealed no significant difference between SMR and GTR with regard to the incidence of postoperative CSF leaks (OR: 1.54; 95% CI: 0.16–15.19, *p* = 0.71). Figure 6 shows a forest plot displaying the results of the analysis. (No significant heterogeneity was present (I^2^ = 0%, *p* = 0.47)).

### 3.6. Publication Bias

To ensure scientific reliability, we undertook the following three steps to analyze any publication bias: first, we conducted an extensive literature search strategy; second, we strictly selected studies for inclusion in this meta-analysis based on the inclusion and exclusion criteria; and finally, we evaluated the publication bias using funnel plots (Figure 7 and Figure 8) and statistical tests for the endpoints (PFS, postoperative meningitis, postoperative intracranial hemorrhage, and postoperative CSF leaks). The data points were all located inside the inverted funnel, indicating a small publication bias in the analysis of the aforementioned endpoints.

Subsequently, Begg’s tests were performed to rule out a publication bias for the applied methods determining the impact of SMR on PFS. As far as PFS analyses according to the multivariate Cox regression analyses were concerned, Begg’s test showed no statistically significant publication bias (Kendall’s tau = −1.0, *p* = 0.32). Moreover, Begg’s test showed no significant publication bias with regard to the analysis of PFS using the mean differences of median PFS time (Kendall’s tau = 0.43, *p* = 0.18).

Begg’s test showed no significant publication bias with regard to the endpoint “postoperative meningitis” (Kendall’s tau = 1.0, *p* = 0.12). As far as postoperative intracranial hemorrhage is concerned, Begg’s test revealed no publication bias (Kendall’s Tau = −1.0, *p* = 0.32). Moreover, Begg’s test identified no publication bias regarding the clinical endpoint “postoperative CSF leak” (Kendall’s tau = −1.0, *p* = 0.32).

## 4. Discussion

In this meta-analysis, we evaluated the effect of supramarginal resection compared to gross total resection of glioblastoma on progression-free survival and postoperative surgical complications in patients with GB. The results of the present meta-analysis were based on the analysis of the following supramarginal resection techniques: resection with a margin of at least 1 cm surrounding the Gd-enhancing tumor in the normal white matter and overlying cortex [24,25], lobectomy [26,28,29], resection of the FLAIR signal alterations if deemed possible [27,30], and resection of the involved entire gyrus [31].

Our results can be summarized as follows: (1) SMR significantly improves progression-free survival in GB patients compared to GTR; (2) SMR does not appear to increase the rate of postoperative meningitis compared to GTR; (3) the incidence of postoperative intracranial hemorrhage is similar among GB patients who underwent SMR or GTR; and (4) the extent of resection (SMR vs. GTR) has no impact on the incidence of postoperative CSF leaks.

### 4.1. Supramarginal Resection and Postoperative Complications

SMR for GB is a promising strategy in maximum cytoreductive neuro-oncological approaches that aim to improve PFS and OS. Those more aggressive surgical therapy avenues are suggested to be associated with an increased risk of perioperative complications. Perioperative complications can delay or interrupt adjuvant treatment or even lead to an omission [35]. Perioperative complications were thereby found to significantly decrease the survival time [36]. In the present meta-analysis, we aimed to compare GTR and SMR in terms of the incidences of the following postoperative surgical adverse events: meningitis, intracranial hemorrhage, and CSF leaks. Those adverse events were reported in only a limited number of studies. However, statistical analysis showed that SMR and GTR have similar incidences regarding the defined surgical complications. This finding is of paramount importance because postoperative complications after surgery for high-grade gliomas were found to significantly worsen the long-term quality of life and perioperative mortality [37,38]. The preservation of a comparable risk profile compared to a conventional GTR may be due to a reduced risk of a postoperative perilesional brain edema in SMR [39]. Furthermore, SMR might even result in a reduced risk of postoperative intracranial hemorrhage because postoperative intralesional bleeding can be found in subtotally or partially removed residual tumor tissue [40].

### 4.2. Supramarginal Resection and Probability of Progression-Free Survival

The extent of resection has been found to significantly impact overall survival time in GB [12,41]. This knowledge is predominantly based on comparisons between gross total resection and subtotal resection. To date, there is no high-class evidence (Level I or Level II) supporting SMR in terms of overall survival or progression-free survival in GB. Previous systematic reviews and meta-analyses have shown that SMR could be associated with improved overall survival [13,15,42], but these studies did not analyze local tumor control, which is better reflected by the probability of PFS.

The present meta-analysis identified that SMR results in enhanced progression-free survival time compared to conventional GTR in GB. Six [24,26,27,28,29,31] of the eight included studies with available PFS data reported that SMR was superior to GTR.

One major SMR technique is the additional resection of FLAIR areas. Mampre et al. [30] performed a conventional resection of the contrast-enhancing areas, and an additional FLAIR resection was performed if it was feasible without causing iatrogenic deficits. However, the additional FLAIR resection or the residual FLAIR volume was not found to be significantly associated with the risk of progression. Additionally, the conventional GTR group and the SMR group had the same median overall survival time (14.9 months). On the other hand, Pessina et al. [27] found that patients who underwent SMR with a 100 % resection of the FLAIR had a median PFS time of 24.5 months, whereas those who underwent a conventional GTR of the contrast-enhancing portions had a median time to PFS of 11.9 months. Further analysis regarding an optimal threshold for OS revealed that the optimum cut-off of 45% additional FLAIR removal enables a significant advantage. Li et al. [32] also found that additional FLAIR removal is superior to conventional GTR regarding median OS time (SMR vs. GTR: 20.7 vs. 15.5 months). Similarly, they identified a threshold regarding additional FLAIR removal at 53.21%. The rates of motor deficits in completely resected tumors were significantly higher in patients with a FLAIR resection of <53.21%. Nevertheless, the rates of other new postoperative neurological complications (e.g., speech impairment, visual impairment, seizure, cognitive/memory status, and sensory deficits) were not significantly different among the completely resected tumors with <53.21% or ≥53.21% resection of the surrounding FLAIR abnormality. In contrast, the studies by Tripathi et al. [10] and Vivas-Buitrago et al. [43] identified lower optimum thresholds ranging from 10 to 20% and at 20% of an additional resection of FLAIR abnormalities surrounding the contrast-enhancing portion regarding the improvement of OS. Furthermore, the study by Vivas-Buitrago et al. [43] also observed that a significant benefit with regard to PFS can be found in patients who underwent a resection of 20 to 40% of the surrounding FLAIR abnormality beyond the conventional GTR of the contrast-enhancing portion, but no significant influence has been shown in patients who underwent an additional FLAIR resection of 50% or greater. All in all, further data are still needed to identify an optimum cut-off value regarding SMR techniques guided by the FLAIR sequence. To date, there are different identified cut-off values for different clinical endpoints (PFS and OS), and not all studies compare their FLAIR-guided SMR techniques to the conventional GTR of the T1-gadolinium-enhancing tumor portion.

Lobectomy is another SMR technique that was applied in three of the eight studies reporting PFS times [26,28,29]. All studies using lobectomies as the SMR technique described superior significant probabilities of PFS compared to conventional GTR of the contrast-enhancing portions. The maximum median PFS time (30.7 months) was observed in the SMR arm in the study by Roh et al. [28]. Furthermore, these studies were comparable in terms of the presence of an IDH-1 mutation. Only the study by Shah et al. [29] included one patient with an IDH-1-mutated GB in the SMR arm, while all other patients in the three trials had an IDH-1 wild-type GB, which would be still classified as WHO grade 4 tumors according to the current WHO classification system [1]. Additionally, all lobectomy studies also demonstrated that this technique is superior regarding OS for completely resectable, noneloquent GBs. The studies by Shah et al. [29] and Schneider et al. [33] also found that lobectomies have the same postoperative risk profile as conventional GTR in terms of postoperative infections and CSF leaks. Moreover, Roh et al. [28] found that the mean postoperative Karnofsky performance status scores were not significantly different between GB patients who underwent SMR or GTR. SMR using lobectomy in a sufficiently selected GB cohort may provide a significant benefit for patients with non-eloquent localized tumors [44]. However, it should be noted that temporal lobectomy carries potential neurological risks, such as short- and long-term memory loss, speech impairment, and executive functioning in cases of a GB on the dominant side [45,46]. Nevertheless, seizures are the most common symptom in patients with high-grade gliomas and represent a relevant issue regarding the quality of life [47]. Hence, temporal lobectomy might not only enhance the OS and PFS times but also improve postoperative seizure outcomes in temporal GBs compared with conventional GTR of temporal GBs [48].

The third major technique for SMR is to extend the resection by setting the resection margins at least 1 cm beyond the contrast-enhancing portion [24,25]. De Bonis et al. [25] retrospectively investigated 36 patients who underwent an extended resection (resection margins at least 1 cm apart from the conventional contrast-enhancing border) and compared them to 52 patients who underwent conventional GTR. A trend towards an enhanced probability of PFS was observed (median time: 12 months vs. 9 months, *p* = 0.09). However, they did not find a significant advantage of this SMR technique in terms of the prolongation of OS. Similarly, Glenn et al. [24] also performed SMR by removing at least 1 cm of surrounding brain tissue beyond the T1 contrast-enhancement margin. This retrospective study found that SMR resulted in a median PFS time of 15 months (compared to 7 months in the GTR arm). Additionally, they demonstrated that patients undergoing SMR had a substantially improved median overall survival compared to the GTR group (24 vs. 11 months). However, only nine and seven patients underwent GTR and SMR, respectively. Furthermore, the SMR arm of this study included one IDH-1-mutated high-grade glioma, whereas the GTR arm included only IDH-1 wild-type GBs. Further insights into this interesting SMR technique will be provided by an ongoing phase II randomized controlled trial comparing an extended resection of at least 1 cm into non-enhancing tissue or the nearest non-enhancing sulcal boundary/ventricle wall if these structures are closer than 1 cm with a conventional GTR of the contrast-enhancing regions of the tumor (NCT number: NCT04737577).

Generally, SMR provides an enhanced PFS in GB and seems to be a safe procedure regarding perioperative surgical complications. As far as SMR techniques are concerned, lobectomy constitutes the technique with the highest evidence as far as the number of studies and included patients are concerned.

### 4.3. Limitations

The present meta-analysis has several limitations. First, all studies included in the analysis reported patient data prior to the new WHO classification of central nervous system tumors [1], and an imbalance of molecular characteristics, such as the IDH mutation, cannot be fully excluded in all studies. However, only one IDH-1-mutated GB patient (in the SMR arm of the study by Glenn et al. [24]) was included in the pooled analysis of hazard ratio effect estimates. Regarding MGMT promotor status, both studies in the pooled analysis of the hazard ratio effect estimates performed multivariate Cox regression models with consideration of the MGMT status [24,28]. The second major limitation of the present meta-analysis is that the results are solely based on mostly smaller retrospective studies. Third, the present meta-analysis only addresses postoperative surgical complications and not postoperative neurological functional status or data on health-related quality of life. New postoperative neurological deficits are known to be associated with worsened outcomes regardless of the postoperative residual tumor volume [40]. Therefore, future studies comparing SMR techniques and conventional GTR will have to consider standardized tools to measure postoperative neurological functioning (e.g., NANO scale), and the tumor’s location eloquence (e.g., Sawaya grading) will have to be taken into account for a reliable analysis [49,50]. Although after surgical treatment, conventional radiochemotherapy with alkylating drugs (temozolomide and lomustine) was the standard treatment in all included studies regarding the meta-analysis of PFS, there is still the crucial limitation that the rates of interruption, delay, or termination of adjuvant therapy are not given or stratified by the extent of resection groups. Hence, there is an urgent need for a larger prospective randomized dataset that considers those issues and potential limitations when analyzing PFS and OS after SMR or conventional GTR.

## 5. Conclusions

The present meta-analysis is the first to investigate the use of SMR for GB regarding PFS and postoperative surgical complications. The findings indicate that compared to conventional GTR, SMR significantly improves PFS time. Furthermore, SMR appears to be comparable to GTR in terms of postoperative surgical complications. However, caution should be exercised when interpreting these results due to the low quality of evidence and lack of a general consensus regarding the definition of SMR. These findings may provide guidance for future prospective randomized trials comparing specific SMR techniques with conventional GTR.

## Figures and Tables

**Figure 1 cancers-15-01772-f001:**
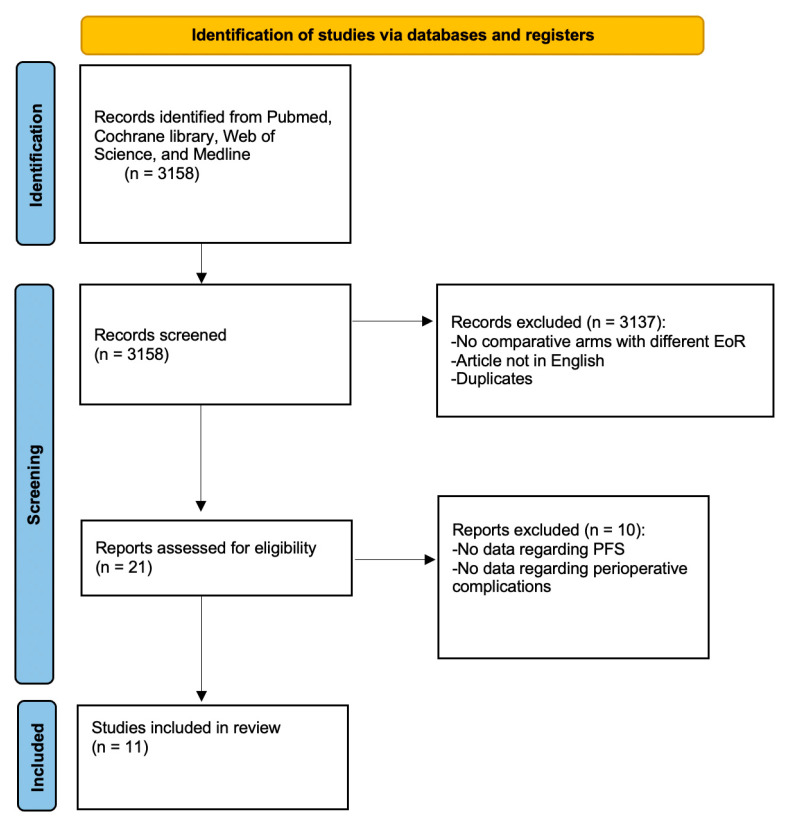
PRISMA flow chart illustrating the study selection.

**Figure 2 cancers-15-01772-f002:**
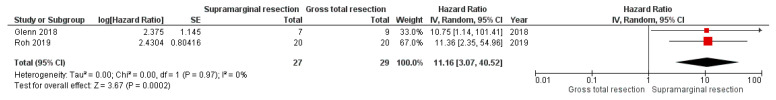
Forest plot displaying log (hazard ratio), HR, and 95% CI estimates for PFS in studies [24,28] evaluating SMR compared to GTR in GB patients. X-axis locations of squares represent the hazard ratio; the bigger the square, the greater the weight due to sample size. The diamond corresponds to the hazard ratio of the overall data.

**Figure 3 cancers-15-01772-f003:**
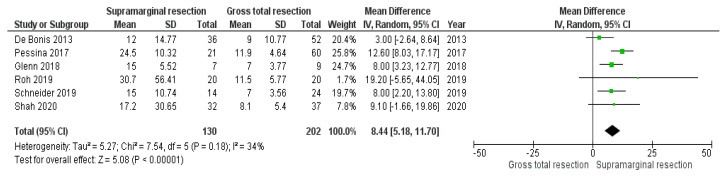
Forest plot with median PFS times and their differences between SMR and GTR [24,25,26,27,28,29]. X-axis locations of squares represent the mean difference; the bigger the square, the greater the weight due to sample size. The diamond corresponds to the mean difference of the overall data.

**Figure 4 cancers-15-01772-f004:**
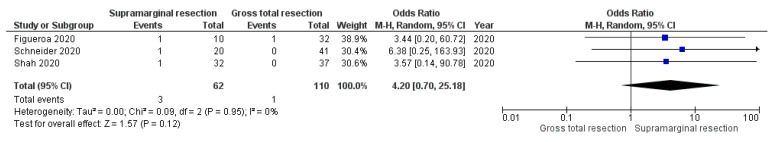
Forest plot displaying odds ratio and 95% CI estimates for postoperative meningitis in studies [29,33,34] evaluating SMR compared to GTR in GB patients. X-axis locations of squares represent the odds ratio; the bigger the square, the greater the weight due to sample size. The diamond corresponds to the odds ratio of the overall data.

**Figure 5 cancers-15-01772-f005:**
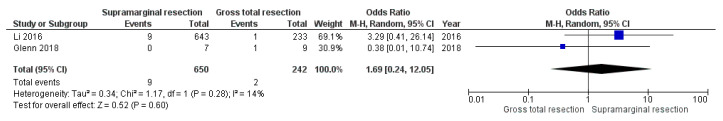
Forest plot displaying odds ratio and 95% CI estimates for postoperative intracranial hemorrhage in studies [24,32] evaluating SMR compared to GTR in GB patients. X-axis locations of squares represent the odds ratio; the bigger the square, the greater the weight due to sample size. The diamond corresponds to the odds ratio of the overall data.

**Figure 6 cancers-15-01772-f006:**
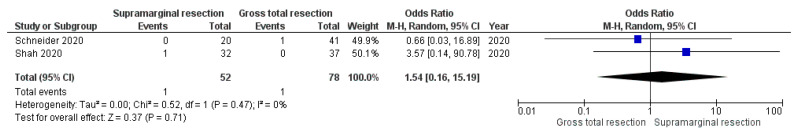
Forest plot displaying odds ratio and 95% CI estimates for postoperative CSF leaks in studies [29,33] evaluating SMR compared to GTR in GB patients. X-axis locations of squares represent the odds ratio; the bigger the square, the greater the weight due to sample size. The diamond corresponds to the odds ratio of the overall data.

**Figure 7 cancers-15-01772-f007:**
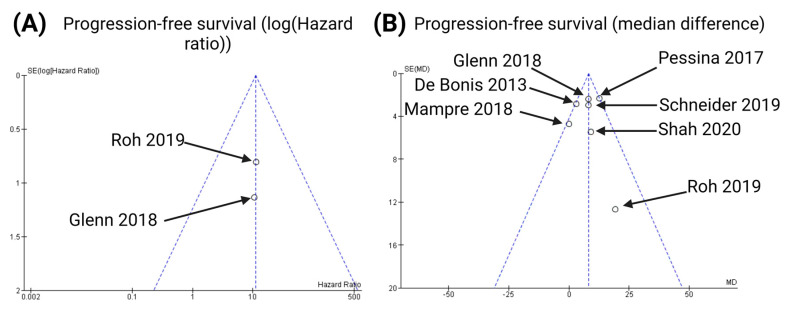
Funnel plots for the following endpoints of the present meta-analysis [24,25,26,27,28,29,30]: probability of progression-free survival according to the multivariate Cox regression analyses (**A**) and probability of progression-free survival according to the mean differences of the median progression-free survival times (**B**).

**Figure 8 cancers-15-01772-f008:**
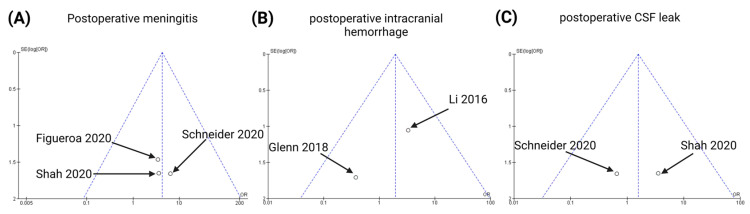
Funnel plots for the following endpoints of the present meta-analysis [24,29,32,33,34]: postoperative meningitis (**A**), postoperative intracranial hemorrhage (**B**), and postoperative CSF leaks (**C**).

**Table 1 cancers-15-01772-t001:** Summary of systematic review of studies comparing supramarginal and gross total resection in GB regarding progression-free survival.

Study	Country	Hierarchy of Evidence	Study Design	Number of Patients	Supramarginal Resection	Gross Total Resection	Age	Female: Male Ratio	Completeness of Adjuvant Radiochemotherapy	Duration of Follow-Up
Glenn et al., 2018 [24]	USA	IV	Retrospective	32 (16 patients with subtotal resection)	7	9	SMR: 56.3 GTR: 58.1	SMR: 1:6 GTR: 1:3.5	NA	NA
De Bonis et al., 2013 [25]	Italy	IV	Retrospective	88	36	52	57.5 (mean age in entire cohort)	1:1.15 (in entire cohort)	NA	NA
Schneider et al., 2019 [26]	Germany	IV	Retrospective	38	14	24	SMR: 63 (mean) GTR: 68 (mean)	SMR: 1:2.5 GTR: 1:2.18	NA	NA
Pessina et al., 2017 [27]	Italy	IV	Retrospective	282 (143 patients with subtotal resection and 58 patients with biopsy)	21	60	61 (median age in entire cohort)	1:1.69 (in entire cohort)	Interruption of chemotherapy in 6 patients (2.1%) and delay/reduction in 14 patients (5.0%)	13.8 months (median f/u for entire cohort)
Roh et al., [28] 2019	Republic of Korea	IV	Retrospective	40	20	20	SMR: 62 (median) GTR: 60 (median)	SMR: 1:2.33 GTR: 1:1.86	NA	46.1 months (median)
Shah et al., 2020 [29]	USA	IV	Retrospective	69	32	37	SMR: 60 (median) GTR: 65 (median)	SMR: 1:3.6 GTR: 1:0.68	61 (89.7%) completed radiochemotherapy. Not stratified by EoR and no data regarding interruption or delay	SMR: 12.4 (median) GTR: 6 (median)
Mampre et al., 2018 [30]	USA	IV	Retrospective	245 (161 patients with subtotal resection)	11	84	59.8 (mean)	1:1.55	Chemotherapy only in 16 (7%) patients or radiation only in 8 (3%) patients. Not stratified by EoR and no data regarding interruption or delay	12.1 (median time for all surviving patients)
Hamada et al., 2016 [31]	Egypt	IV	Prospective	59 (14 patients with subtotal resection and 4 patients with debulking)	20	21	48.57 (mean)	1:2.69	NA	NA

Abbreviations: EoR = extent of resection; GB = glioblastoma; GTR = gross total resection; NA = not available; SMR = supramarginal resection.

**Table 2 cancers-15-01772-t002:** Summary of surgical techniques and progression-free survival analyses in studies comparing supramarginal resection and gross total resection.

Study	Supramarginal Resection Technique	Supramarginal Resection and PFS (Months)	Gross Total Resection and PFS (Months)	Method of Statistical Comparison	Available Multivariate Statistical Results	Study Limitations
Glenn et al., 2018 [24]	Resection extended beyond the T1 contrast-enhancement margin to include at least 1 cm of surrounding brain. Tumors with T1 contrast-enhancement less than 1 cm from the temporal cortex were included in the supramaximal resection group when the resection included the overlying cortex, as well as at least a 1 cm brain margin in all other directions.	15 (median)	7 (median)	Multivariate Cox proportional hazard model	Cox regression (including hazard ratios, 95% CI, *p*-values)	Retrospective design
De Bonis et al., 2013 [25]	Extent of resection was classified into two groups: “border resection” (resection margins at the level of tumor border (= contrast-enhanced peripheral areas of tumors) or “extended resection” (ER, resection margins beyond tumor borders, i.e., in the apparently normal white matter, 1–2 cm far from tumor border).	12 (median)	11 (median)	Log-rank test	Not available	Retrospective design, no multivariate Cox regression analysis of PFS
Schneider et al., 2019 [26]	Gross total resection of contrast-enhancing tumor portion of temporal GB was compared with patients who underwent temporal tumor resection with additional anterior temporal lobectomy (from the temporal tip to posterior margin of resection at nondominant side: 5–6 cm and 4–5 cm on the dominant hemisphere	15 (median)	7 (median)	Log-rank test, multivariate logistic regression analysis	Multivariate logistic regression analysis	Retrospective design, no multivariate Cox regression analysis of PFS
Pessina et al., 2017 [27]	SMR was defined as surgical resection of 100% of contrast-enhanced and 100% of FLAIR-altered tumor areas.	24.5 (median)	11.9 (median)	Log-rank test, *p*-values of multivariate Cox model	Multivariate Cox regression analysis (only *p*-values available)	Retrospective design, no hazard ratios or confidence intervals of multivariate Cox regression analysis
Roh et al., 2019 [28]	SMR: temporal lobectomy for temporal GB with the posterior margin of resection approximately 5–6 cm from the temporal pole. Anterior portion of superior temporal gyrus was also removed. Frontal lobectomy was performed for frontal GB. Corpus callosum was resected if it was invaded. The posterior margin of frontal lobectomy was just beneath the coronal suture, which is considered to be 1–2 cm anterior to the precentral sulcus.	30.7 (median)	11.5 (median)	Log-rank test, multivariate Cox model	Multivariate Cox regression (including hazard ratios, 95% CI, *p*-values)	Retrospective design
Shah et al., 2020 [29]	SMR: temporal lobectomy for temporal GB with the posterior margin of resection approximately 5–6 cm from the temporal pole. Anterior portion of superior temporal gyrus was also removed. Frontal lobectomy was performed for frontal GB. Corpus callosum was resected if it was invaded. The posterior margin of frontal lobectomy was just beneath the coronal suture, which is considered to be 1–2 cm anterior to the precentral sulcus. Occipital lobectomies were also performed.	17.2 (median)	8.1 (median)	Log-rank test, multivariate Cox regression	Multivariate Cox regression model (only *p*-values available)	Retrospective design, only *p*-values of Cox regression available
Mampre et al., 2018 [30]	FLAIR resection was performed if resection was possible without causing iatrogenic deficits.	NA	NA	Log-rank tests, multivariate Cox regression analysis	Multivariate Cox regression analysis (including hazard ratios, 95% CI, *p*-values)	Retrospective design; postoperative FLAIR volume and no extent of resection was analyzed regarding PFS. No mean or median times to PFS stratified by GTR and SMR
Hamada et al., 2016 [31]	Anatomical resection (AR) of the involved entire gyrus was performed if it was classified as noneloquent.	Frontal AR: 10.75 (mean) Occipital AR: 7.5 (mean) Parietal AR: not performed Temporal AR: 12.25 (mean)	Frontal GTR: 8.5 (mean) Occipital GTR: 6 (mean) Parietal GTR: 4.67 (mean) Temporal GTR: 9.43 (mean)	Not available	Not available	No statistical comparison of EoR regarding PFS

Abbreviations: GB = glioblastoma; GTR = gross total resection; NA = not available; PFS = progression-free survival; SMR = supramarginal resection.

**Table 3 cancers-15-01772-t003:** Perioperative complications in studies comparing supramarginal resection with gross total resection of glioblastoma.

Study	Mortality	Meningitis	Intracranial Hemorrhage	CSF Leak
Li et al., 2016 [32]	NA	NA	SMR: 9/643 GTR: 1/233	NA
Glenn et al., 2018 [24]	NA	SMR: 0/7 GTR: 0/9	SMR: 0/7 GTR: 1/9	NA
Schneider et al., 2020 [33]	NA	SMR: 1/20 GTR: 0/41	NA	SMR: 0/20 GTR: 1/41
Pessina et al., 2017 [27]	SMR: 0/21 GTR: 0/60	NA	NA	NA
Shah et al., 2020 [29]	NA	SMR: 1/32 GTR: 0/37	NA	SMR: 1/32 GTR: 0/37
Figueroa et al., 2020 [34]	NA	SMR: 1/10 GTR: 1/32	NA	1 CSF leak (not stratified by EoR)
Hamada et al., 2016 [31]	SMR: 0/16 GTR: 0/16	NA	NA	NA

Abbreviations: CSF = cerebrospinal fluid; EoR = extent of resection; GTR = gross total resection; NA = not available; SMR = supramarginal resection.

## Data Availability

All data were included in this manuscript.

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
