# Peer review of "A Systematic Review and Meta-Analysis of Supramarginal Resection versus Gross Total Resection in Glioblastoma: Can We Enhance Progression-Free Survival Time and Preserve Postoperative Safety?"

_cancers, 2023, doi:10.3390/cancers15061772_

Round 1
Reviewer 1 Report
Authors summarized and analyzed about effect of supramarginal resection in glioblastoma. Recently, several authors reported that FLAIRectomy prolonged PFS and OS without decreasing KPS. Your study is important for glioma surgeon. However, you have to analyze more.
1. About complication: For glioma surgery, CSF leakage and meningitis are problem for craniotomy, location of tumor and postoperative management. And postoperative hemorrhage is occurred by character of tumor and surgical technique. Most important complication is neurological disorder, and we have to resect tumor without worsening for FLAIRectomy.
You must refer complication about neurological disorder (new deficit, worsening, permanent or temporary, and KPS).
2. You summarize and analyze various resection rate all together. Many neurosurgeon know lobectomy improves OS. How much FLAIR area do we resect to improve PFS?
Author Response
Dear Reviewer
Thank you for reading our manuscript and critically reviewing it, which will help us improve it to a better scientific level and make it more understandable to the readership.
In the following we would like to respond to your remarks:
We agree with the reviewer that FLAIRectomy is an emerging surgical technique in the field of supramarginal resections (SMR) and the potential benefit of enhanced overall survival (OS) / progression-free survival (PFS) must be tempered by the effect that new postoperative neurological deficits are associated with worsened outcomes regardless of the postoperative residual tumor volume (1). Five studies describe FLAIRectomy for high-grade gliomas and report about various endpoints (OS, PFS, KPS) (2-6). First, Tripathi et al. (2) reports that the postoperative KPS scores were not inversely associated with increasing percentages of FLAIRectomy. However, this finding reflects the interindividual comparison between different degrees of FLAIRectomy and was not compared with a conventional gross total resection (GTR). Data regarding neurological deficits and PFS are not given in this study. However, it was found that a significant benefit in OS was seen with thresholds from 10% to 20% regarding FLAIRectomy. Second, Vivas-Buitrago et al. (3) identified a very similar optimum cut-off value regarding OS (20%). As far as PFS is concerned, this retrospective study identified that a significant benefit can be observed in patients who underwent the resection of 20% to 40% of the surrounding FLAIR abnormality beyond the conventional GTR of the contrast-enhancing portion, but no significant impact was seen in patients who underwent a FLAIRectomy of 50% or greater. Furthermore, they found that the postoperative KPS did not differ between patients who underwent a conventional GTR or an additional FLAIRectomy (80 vs. 80, p = 0.15). However, the KPS data cannot be extracted for a statistical analysis because no range or confidence interval of the given mean KPS values are provided. Furthermore, detailed data regarding neurological deficits are also not given in this study. Third, Li et al. (3) retrospectively reviewed the survival data of 1229 patients with glioblastoma (GB). They identified that the resection of ≥53.21% of the surrounding FLAIR abnormality beyond the 100% contrast-enhancing resection was associated with prolonged survival. Nevertheless, PFS was not analyzed in this study. Postoperative KPS course stratified by FLAIRectomy or GTR is also not given. Surprisingly, they found in completely resected tumors that the rates of motor deficits were significantly higher in patients with FLAIR resection < 53.21% compared to those with a FLAIR resection ≥ 53.21%. The rates of other neurological complications (e.g., speech impairment, visual impairment, seizure, cognitive/memory status, sensory deficits) were not significantly different among the completely resected tumors with <53.21% or ≥53.21% resection of the surrounding FLAIR abnormality. However, these rates were not compared to those who underwent only a conventional gross total resection of the contrast-enhancing tumor portion only. Furthermore, this is only a univariate result without the consideration of the tumor´s location eloquence. Forth, Certo et al. (5) prospectively investigated 68 GB patients regarding the association between extent of resection (EOR) based on FLAIR with OS and PFS. They identified a significant correlation between EOR and OS as well as PFS. Furthermore, it was reported that no statistically significant differences in KPS variations were observed during follow-up. However, KPS as well as PFS/OS are not stratified with absolute data for a comparison between GTR and FLAIRectomy. Neurological deficits were also not stratified by extent of resection based on T1-Gd-enhanced sequences or FLAIR. Fifth, Mampre et al. (6) retrospectively reviewed 245 patients regarding FLAIR resection and conventional GTR. Postoperative FLAIR volume was not found to be associated with overall survival or progression-free survival. They found that the mean percent of FLAIR resection did not differ significantly among those with or without new postoperative neurological deficits. Nevertheless, a comparison of postoperative KPS or neurological deficits comparing GTR with SMR by removing surrounding FLAIR abnormalities was not performed. All in all, further data are still needed to identify an optimum cut-off value regarding SMR techniques guided by the FLAIR sequence. To date, there are different optimum cut-off values for different clinical endpoints (PFS & OS) and not all these studies compare the FLAIR-guided SMR techniques to the conventional GTR of the T1-gadolinium enhancing tumor portion. Therefore, we have revised our discussion about this very important topic to further enhance the understanding of this issue to the readership. Unfortunately, during the revision process of our study, we identified an error in our data extraction of PFS times from the study by Mampre et al. (6). Inadvertently the median time to OS (stratified by conventional GTR & SMR) was extracted as the median time to PFS. Median time to progression stratified by conventional GTR or SMR is not given in this study. Hence, we revised line 35 in the abstract, the section “3.3“, and the section “3.6”. The corresponding Figure 3 and Figure 7 (B) demonstrating the forest plot and funnel plot summarizing the probabilities of progression-free survival according to the mean differences of the median progression-free survival times in patients who underwent either conventional GTR or SMR were also revised. Nevertheless, the key finding did not change after the revision of this mistake and the median time to PFS is still significantly longer in those who underwent a SMR. The median PFS time of the SMR arm was 8.44 months (95%CI:5.18-11.70, p<0.00001) months longer than the GTR arm. We sincerely apologize for this mistake. We absolutely agree with the reviewer regarding the potential benefit of FLAIRectomy in the field of supramarginal resections. Unfortunately, the present literature provides not enough data to allow reliable statistical analyses comparing conventional GTR with FLAIRectomy in terms of KPS and new postoperative neurological deficits. However, we newly created the supplementary table 1 (see supplementary material), which summarizes the current data regarding postoperative KPS and new neurological deficits in all studies comparing SMR techniques with conventional GTR. Postoperative KPS values are provided in only 6 studies (7-12) and only 4 (8-11) of those studies further stratified the postoperative KPS values by EOR. The study by Schneider et al. (8) revealed that patients who underwent a temporal lobectomy had a significantly superior KPS at 12-months after surgery compared with patients who underwent a conventional GTR. The other three studies comparing postoperative KPS values stratified by EOR found no significant difference. Statistical analysis of the postoperative KPS was not possible because different statistical values (mean or median values with range or interquartile range) at different timepoints are given. The rates of new postoperative neurological deficits are described in 7 studies (4, 6, 11-15) and three studies describe the incidence of new postoperative deficits in patients who underwent either SMR or GTR (11, 13, 14). No significant associations between SMR or GTR with the onset of new postoperative neurological deficits were found. Statistical analysis of new postoperative deficits was not possible because only one event in the study by Glenn et al. (13) was observed and the other studies comparing new deficits stratified by SMR or GTR observed no events (11, 14). Future prospective randomized trials comparing SMR techniques with GTR of T1-contrast-enhanced tumor portion will have to consider the NANO scale to sufficiently address the perioperative neurological functioning in GB patients and the tumor´s location eloquence (e.g., Sawaya grading) also has to be considered to reliably compare SMR techniques such as the FLAIRectomy with conventional GTR of the Gd-enhancing tumor lesion (16, 17).
References
- McGirt MJ, Mukherjee D, Chaichana KL, Than KD, Weingart JD, Quinones-Hinojosa A. Association of surgically acquired motor and language deficits on overall survival after resection of glioblastoma multiforme. Neurosurgery.2009 Sep;65(3):463-9; discussion 469-70. doi: 10.1227/01.NEU.0000349763.42238.E9.
- Tripathi S, Vivas-Buitrago T, Domingo RA, Biase G, Brown D, Akinduro OO, Ramos-Fresnedo A, Sherman W, Gupta V, Middlebrooks EH, Sabsevitz DS, Porter AB, Uhm JH, Bendok BR, Parney I, Meyer FB, Chaichana KL, Swanson KR, Quiñones-Hinojosa A. IDH-wild-type glioblastoma cell density and infiltration distribution influence on supramarginal resection and its impact on overall survival: a mathematical model. J Neurosurg. 2021 Oct 29:1-9. doi: 10.3171/2021.6.JNS21925.
- Vivas-Buitrago T, Domingo RA, Tripathi S, De Biase G, Brown D, Akinduro OO, Ramos-Fresnedo A, Sabsevitz DS, Bendok BR, Sherman W, Parney IF, Jentoft ME, Middlebrooks EH, Meyer FB, Chaichana KL, Quinones-Hinojosa A. Influence of supramarginal resection on survival outcomes after gross-total resection of IDH-wild-type glioblastoma. J Neurosurg. 2021 Jun 4;136(1):1-8. doi: 10.3171/2020.10.JNS203366.
- Li YM, Suki D, Hess K, Sawaya R. The influence of maximum safe resection of glioblastoma on survival in 1229 patients: Can we do better than gross-total resection? J Neurosurg. 2016 Apr;124(4):977-88. doi: 10.3171/2015.5.JNS142087.
- Certo F, Altieri R, Maione M, Schonauer C, Sortino G, Fiumanò G, Tirrò E, Massimino M, Broggi G, Vigneri P, Magro G, Visocchi M, Barbagallo GMV. FLAIRectomy in Supramarginal Resection of Glioblastoma Correlates With Clinical Outcome and Survival Analysis: A Prospective, Single Institution, Case Series. Oper Neurosurg (Hagerstown). 2021 Jan 13;20(2):151-163. doi: 10.1093/ons/opaa293.
- Mampre D, Ehresman J, Pinilla-Monsalve G, Osorio MAG, Olivi A, Quinones-Hinojosa A, Chaichana KL. Extending the resection beyond the contrast-enhancement for glioblastoma: feasibility, efficacy, and outcomes. Br J Neurosurg. 2018 Oct;32(5):528-535. doi: 10.1080/02688697.2018.1498450.
- De Bonis P, Anile C, Pompucci A, Fiorentino A, Balducci M, Chiesa S, Lauriola L, Maira G, Mangiola A. The influence of surgery on recurrence pattern of glioblastoma. Clin Neurol Neurosurg. 2013 Jan;115(1):37-43. doi: 10.1016/j.clineuro.2012.04.005.
- Schneider M, Potthoff AL, Keil VC, Güresir Á, Weller J, Borger V, Hamed M, Waha A, Vatter H, Güresir E, Herrlinger U, Schuss P. Surgery for temporal glioblastoma: lobectomy outranks oncosurgical-based gross-total resection. J Neurooncol. 2019 Oct;145(1):143-150. doi: 10.1007/s11060-019-03281-1.
- Roh TH, Kang SG, Moon JH, Sung KS, Park HH, Kim SH, Kim EH, Hong CK, Suh CO, Chang JH. Survival benefit of lobectomy over gross-total resection without lobectomy in cases of glioblastoma in the noneloquent area: a retrospective study. J Neurosurg. 2019 Mar 1;132(3):895-901. doi: 10.3171/2018.12.JNS182558.
- Shah AH, Mahavadi A, Di L, Sanjurjo A, Eichberg DG, Borowy V, Figueroa J, Luther E, de la Fuente MI, Semonche A, Ivan ME, Komotar RJ. Survival benefit of lobectomy for glioblastoma: moving towards radical supramaximal resection. J Neurooncol. 2020 Jul;148(3):501-508. doi: 10.1007/s11060-020-03541-5.
- Schneider M, Ilic I, Potthoff AL, Hamed M, Schäfer N, Velten M, Güresir E, Herrlinger U, Borger V, Vatter H, Schuss P. Safety metric profiling in surgery for temporal glioblastoma: lobectomy as a supra-total resection regime preserves perioperative standard quality rates. J Neurooncol. 2020 Sep;149(3):455-461. doi: 10.1007/s11060-020-03629-y.
- Figueroa J, Morell A, Bowory V, Shah AH, Eichberg D, Buttrick SS, Richardson A, Sarkiss C, Ivan ME, Komotar RJ. Minimally invasive keyhole temporal lobectomy approach for supramaximal glioma resection: A safety and feasibility study. J Clin Neurosci. 2020 Feb;72:57-62. doi: 10.1016/j.jocn.2020.01.031.
- Glenn CA, Baker CM, Conner AK, Burks JD, Bonney PA, Briggs RG, Smitherman AD, Battiste JD, Sughrue ME. An Examination of the Role of Supramaximal Resection of Temporal Lobe Glioblastoma Multiforme. World Neurosurg. 2018 Jun;114:e747-e755. doi: 10.1016/j.wneu.2018.03.072
- Pessina F, Navarria P, Cozzi L, Ascolese AM, Simonelli M, Santoro A, Clerici E, Rossi M, Scorsetti M, Bello L. Maximize surgical resection beyond contrast-enhancing boundaries in newly diagnosed glioblastoma multiforme: is it useful and safe? A single institution retrospective experience. J Neurooncol. 2017 Oct;135(1):129-139. doi: 10.1007/s11060-017-2559-9.
- Hamada SM, Ahmed HA. Anatomical resection in glioblastoma: extent of resection and its impact on duration of survival. Egypt J Neurol Psychiatry Neurosurg. 2016; 53(3): 135-145
- Ung TH, Ney DE, Damek D, Rusthoven CG, Youssef AS, Lillehei KO, Ormond DR. The Neurologic Assessment in Neuro-Oncology (NANO) Scale as an Assessment Tool for Survival in Patients With Primary Glioblastoma. Neurosurgery. 2019 Mar 1;84(3):687-695. doi: 10.1093/neuros/nyy098.
- Sawaya R, Hammoud M, Schoppa D, Hess KR, Wu SZ, Shi WM, Wildrick DM. Neurosurgical outcomes in a modern series of 400 craniotomies for treatment of parenchymal tumors. Neurosurgery. 1998 May;42(5):1044-55; discussion 1055-6. doi: 10.1097/00006123-199805000-00054.
Reviewer 2 Report
The authors have performed a detailed systematic review and meta analysis of the effect of supramarginal resection for hgg. The study is well written and very well conducted. My only comment is that it is unclear to me whether the authors take into account if the patients in both the supramarginal resection group and GTR group received same post op oncological treatment. There might be a difference and if so, it should be noted.
Author Response
Dear Reviewer
Thank you for reading our manuscript and critically reviewing it, which will help us improve it to a better scientific level and make it more understandable to the readership.
In the following we would like to respond to your remarks:
We absolutely agree with the reviewer that the type of postoperative adjuvant oncological treatment is an essential point which is a potential limitation of the present included studies limiting the results of the current meta-analysis. Hence, we revised the section limitation at the end of the discussion and added this essential issue to the limitations of the analysis regarding the endpoint “PFS”. Furthermore, we added the available information regarding completion, delay, interruption, or termination of adjuvant therapies of the studies reporting PFS data as a new column in table 1 (see Section “3.2”). All included studies reporting PFS times for GB patients who underwent SMR or conventional GTR described that after surgical treatment conventional radiochemotherapy with alkylating drugs (temozolomide only or combined with lomustine) was the standard treatment. Nevertheless, two studies report overall data regarding completion, interruption, delay, or reduction of the radiochemotherapy without further stratifying those rates by the extent of resection [1, 2]. The study by Pessina et al. [1] is the only included investigation that describes rates regarding interruption, delay, reduction, or termination of the adjuvant therapies. They described that the chemotherapy was interrupted in 6 patients (2.1%) and delayed or reduced in 14 patients (5.0%). However, those rates are not further stratified by the extent of resection (SMR or GTR) which would have been necessary for a meta-analysis. The study by Shah et al. [2] at least described that 61 (89.7%) out of 68 patients completed postoperative radiochemotherapy. However, for the residual patients it is unclear whether an interruption, termination or reduction of the therapy was performed, and the available data is only given for the whole cohort without a stratification by the extent of resection. Hence, there is the urgent need for a larger prospective randomized dataset that considers those issues and potential limitations when analyzing PFS and OS after SMR or conventional GTR. Unfortunately, during the revision process of our study, we identified an error in our data extraction of PFS times from the study by Mampre et al. (3). Inadvertently the median time to OS (stratified by conventional GTR & SMR) was extracted as the median time to PFS. Median time to progression stratified by conventional GTR or SMR is not given in this study. Hence, we revised line 35 in the abstract, the section “3.3“, and the section “3.6”. The corresponding Figure 3 and Figure 7 (B) demonstrating the forest plot and funnel plot summarizing the probabilities of progression-free survival according to the mean differences of the median progression-free survival times in patients who underwent either conventional GTR or SMR were also revised. Nevertheless, the key finding did not change after the revision of this mistake and the median time to PFS is still significantly longer in those who underwent a SMR. The median PFS time of the SMR arm was 8.44 months (95%CI:5.18-11.70, p<0.00001) months longer than the GTR arm. We sincerely apologize for this mistake.
References
- Pessina, F.; Navarria, P.; Cozzi, L.; Ascolese, A.M.; Simonelli, M.; Santoro, A.; Clerici, E.; Rossi, M.; Scorsetti, M.; Bello, L. Maximize surgical resection beyond contrast-enhancing boundaries in newly diagnosed glioblastoma multiforme: is it useful and safe? A single institution retrospective experience. J Neurooncol. 2017, 135(1), 129-139
- Shah, A.H.; Mahavadi, A.; Di, L.; Sanjurjo, A.; Eichberg, D.G.; Borowy, V.; Figueroa, J.; Luther, E.; de la Fuente, M.I.; Semonche, A.; Ivan, M.E.; Komotar, R.J. Survival benefit of lobectomy for glioblastoma: moving towards radical supramaximal resection. J Neurooncol. 2020, 148(3), 501-508
- Mampre D, Ehresman J, Pinilla-Monsalve G, Osorio MAG, Olivi A, Quinones-Hinojosa A, Chaichana KL. Extending the resection beyond the contrast-enhancement for glioblastoma: feasibility, efficacy, and outcomes. Br J Neurosurg. 2018 Oct;32(5):528-535. doi: 10.1080/02688697.2018.1498450.